# Expanding the genome-targeting scope and the site selectivity of high-precision base editors

Junjie Tan [1], Fei Zhang[1,2], Daniel Karcher[1] & Ralph Bock [1]*

Base editors (BEs) are RNA-guided CRISPR-Cas-derived genome editing tools that induce single-nucleotide changes. The limitations of current BEs lie in their low precision (especially when multiple target nucleotides of the deaminase are present within the activity window) and their restriction to targets that are in proper distance from the PAM sequence. We have recently developed high-precision cytidine BEs by engineering CDA1 truncations and nCas9 fusions that predominantly edit nucleotide $C_{-18}$ relative to the PAM sequence NGG. Here, by testing fusions with Cas9 variants that recognize alternative PAMs, we provide a series of high-precision BEs that greatly expand the versatility of base editing. In addition, we obtained BEs that selectively edit $C_{-15}$ or $C_{-16}$. We also show that our high-precision BEs can substantially reduce off-target effect. These improved base editing tools will be widely applicable in basic research, biotechnology and gene therapy.

[1] Max-Planck-Institut für Molekulare Pflanzenphysiologie, Am Mühlenberg 1, D-14476 Potsdam-Golm, Germany. [2] Present address: National Key Laboratory of Crop Genetic Improvement, Huazhong Agricultural University, Wuhan, Hubei 430070, China. *email: rbock@mpimp-golm.mpg.de

CRISPR-Cas systems provide adaptive immunity to bacteria by protecting their hosts from viruses, plasmids and other types of invasive nucleic acids. The CRISPR RNA confers the site specificity and recognizes the target site through complementary base pairing. Cas proteins are endonucleases typically comprised of two nuclease domains to cleave both strands of the target DNA[1–4]. Several CRISPR-Cas systems, especially the one based on the Cas9 enzyme from *Streptococcus pyogenes* (SpCas9), have been successfully repurposed for genome editing in a wide range of organisms[2,5–7]. Upon repair of the double-strand break by the endogenous DNA repair machinery of the cell, randomly formed insertions or deletions (indels) are generated in the target site, typically resulting in loss-of-function alleles (gene knockouts).

As most of the mutations causing hereditary diseases in humans and much of the useful genetic variation in plant breeding represent point mutations rather than loss-of-function mutations, conventional CRISPR-Cas tools are of limited use in gene therapy and precision breeding[8–11]. Therefore, substantial efforts have been directed towards reengineering CRISPR-Cas systems for site-directed mutagenesis. Homology-directed repair (HDR) stimulated by DSBs can be used to introduce precise changes into target DNA sequences. However, while HDR is efficient is some therapeutically relevant cell types (e.g., T cells and some stem cells[12]), it suffers from low editing efficiency in many other cell types[13]. Moreover, HDR requires the presence of donor DNA as repair template, and has to compete with other DNA repair pathways such as non-homologous end joining (NHEJ), which can produce indels and other undesired mutations[13]. Recently, base editors (BEs) have been developed by converting Cas endonucleases into programmable nucleotide deaminases[14–16] that facilitate the introduction of C-to-T mutations (by cytidine-to-uridine deamination) or A-to-G mutations (by adenosine-to-inosine deamination) without inducing a double-strand break in the target DNA. BEs are widely applicable as universal tools for site-directed mutagenesis in vivo. For example, they can correct disease-causing point mutations in humans or introduce single-nucleotide changes that underlie quantitative trait loci (QTLs) for yield, resistance or food quality[9,17–21].

A key limitation in the applicability of current BEs lies in their relatively wide activity window also referred to as low editing precision[22,23]. For example, cytosine BEs can potentially edit any C within an ~4–17 nt wide window in the protospacer[10,14,15,24]. Unfortunately, many human disease-associated alleles (e.g., the β-thalassemia locus *HBB*, the Alzheimer's disease-associated gene *APOE4*, and the oculocutaneous albinism-related locus *TYR*) have multiple bystander Cs within the activity window[15,22,25]. Their change by the BE would cause unwanted mutations in the target gene, and such off-site mutations would be unacceptable in many practical applications of base editing.

Recently, we developed high-precision BEs by designing optimized nCas9 fusions to cytidine deaminase domains. Testing a series of truncated versions of CDA1, the AID homolog of sea lamprey[14], we identified BEs that preferentially edit position $C_{-18}$ relative to the PAM sequence[23]. However, these BEs still require the presence of the PAM motif NGG in the appropriate distance from the target nucleotide[23]. In the present study, we seek to overcome this serious limitation and provide a more versatile set of high-precision BEs. To this end, we replaced Cas9 by a series of Cas9 variants with altered PAM specificities. Our results show that, when combined with optimized CDA1 truncations, high-precision BEs with altered PAM specificities can be obtained, thus greatly expanding the base editing scope. As the previously described editors for the PAM motif NGG[23], the BEs preferentially edit position $C_{-18}$. By testing other deaminases and

engineering their connection to Cas9, we also obtain high-precision BEs that selectively edit $C_{-15}$ or $C_{-16}$. Together, these tools substantially expand the toolbox for precise gene editing and will enable applications of base editing in reverse genetics, gene therapy, and precision breeding.

## Results

**Expanding precision base editing to non-NGG PAM sequences.** In previous work, we constructed high-precision CDA1-based cytosine BEs that predominantly edit position $C_{-18}$ relative to the PAM sequence[23]. However, the application of these BEs remains largely restricted to the PAM sequence NGG and the presence of the target nucleotide at position −18. Recently, several Cas9 variants have been described that recognize non-NGG PAM sequences[26–28]. To test whether Cas9 variants with expanded PAM compatibility can be used in our high-precision BEs to extend their DNA targeting scope, we replaced the nCas9 sequence with that of four different nCas9 variants recognizing four different non-NGG PAMs (Fig. 1a, b). Of particular interest is the minimal PAM sequence NG (as recognized by variant SpCas9-NG; Fig. 1b), which occurs much more frequently in DNA sequences than the wild-type PAM sequence NGG. As deaminase domain, we tested the full-length CDA1 and a series of truncated CDA1 versions that lack 13–20 C-terminal amino acids. When fused to nCas9, this range of C-terminal deletions was shown previously to provide the maximum increase in editing precision while retaining high editing activity[23]. In this way, 32 BEs were constructed: the full-length CDA1 (as N-terminal or C-terminal fusion) and 6 CDA1 deletions combined with the VQR-Cas9 variant (nCDA1Δ195-VQRBE3; nCDA1Δ194-VQRBE3; nCDA1Δ193-VQRBE3; nCDA1Δ192-VQRBE3; nCDA1Δ190-VQRBE3; nCDA1Δ188-VQRBE3; Fig. 1a, c; Supplementary Fig. 1) that recognizes the PAM sequence NGA (Fig. 1b), the full-length CDA1 (as N-terminal or C-terminal fusion) and 6 CDA1 deletions combined with the VRER-Cas9 variant (nCDA1Δ195-VRERBE3; nCDA1Δ194-VRERBE3; nCDA1Δ193-VRERBE3; nCDA1Δ192-VRERBE3; nCDA1Δ190-VRERBE3; nCDA1Δ188-VRERBE3; Fig. 1d; Supplementary Fig. 2) that recognizes the PAM sequence NGCG (Fig. 1b), the full-length CDA1 (as N-terminal or C-terminal fusion) and 6 CDA1 deletions combined with the xCas9 variant (nCDA1Δ195-xBE3; nCDA1Δ194-xBE3; nCDA1Δ193-xBE3; nCDA1Δ192-xBE3; nCDA1Δ190-xBE3; nCDA1Δ188-xBE3; Fig. 1e; Supplementary Fig. 3) that recognizes the PAM sequences NG, GAA, and GAT (Fig. 1b), and the full-length CDA1 (as N-terminal or C-terminal fusion) and 6 CDA1 deletions combined with the SpCas9-NG variant (nCDA1Δ195-NGBE3; nCDA1Δ194-NGBE3; nCDA1Δ193-NGBE3; nCDA1Δ192-NGBE3; nCDA1Δ190-NGBE3; nCDA1Δ188-NGBE3; Fig. 1f, g; Supplementary Fig. 4) that recognizes the PAM sequence NG (Fig. 1b).

For each set of BEs, we tested target sites that contain a stretch of consecutive cytidines within the activity window upstream of the PAM (Supplementary Table 1). PolyC motifs were used to provide the most rigorous test for editing precision, in that specific editing of a single C would require maximum discriminatory power. Editing efficiency and precision were first assessed by dideoxy chain termination sequencing of amplified PCR products (Supplementary Figs. 1–4), and the two best-performing BEs were then further characterized by high-throughput next-generation sequencing (Fig. 1; see Methods; ref.[23]).

The VQR-Cas9 variant recognizes the PAM sequence NGA (Fig. 1b). As expected, the full-length VQR-Cas9 BE (nCDA1-VQRBE3) edited with low precision in a larger window upstream of

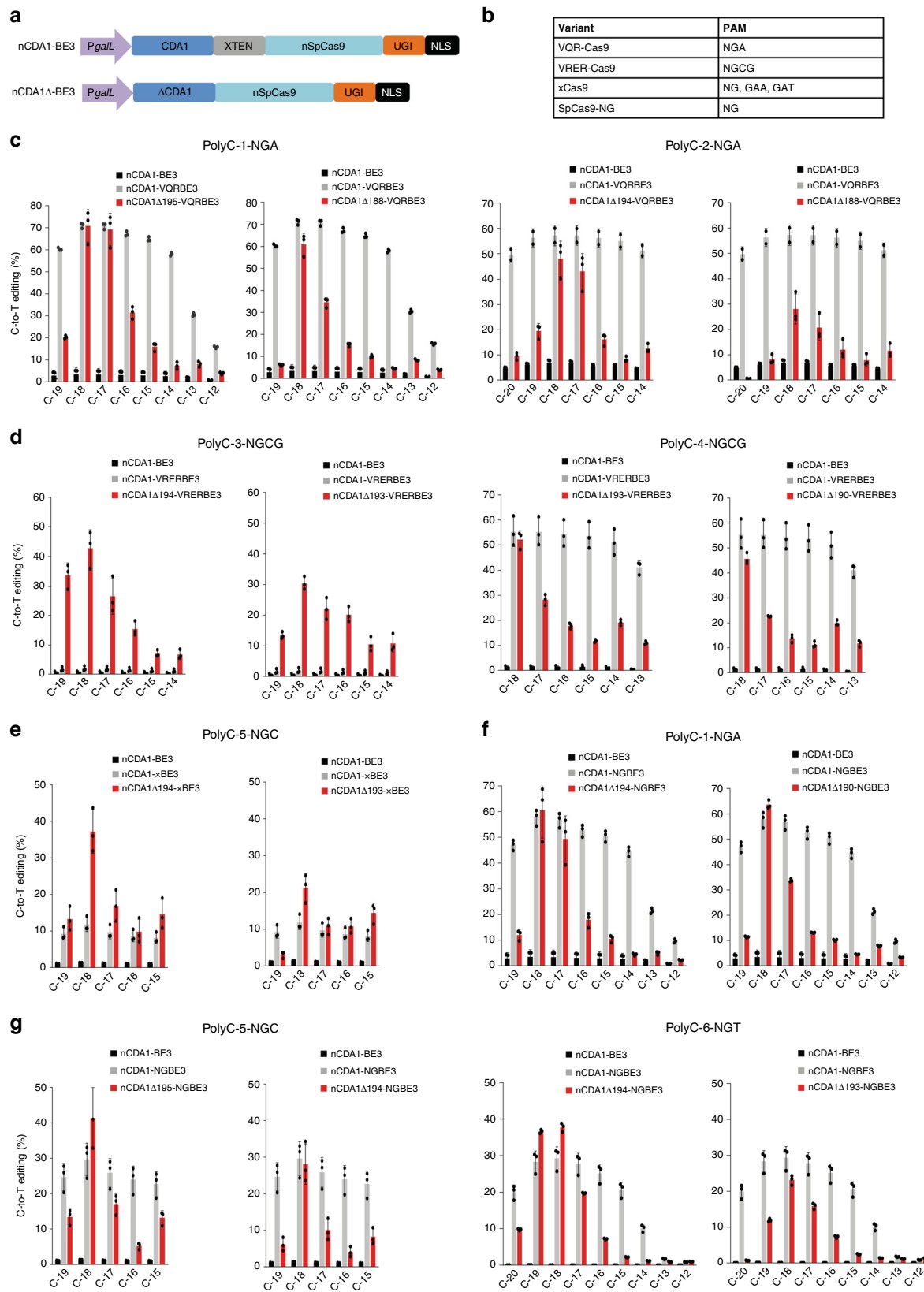

the PAM (Supplementary Fig. 1). The activity window ranged from $C_{-14}$ to $C_{-19}$ in target sequence PolyC-1-NGA and from $C_{-14}$ to $C_{-20}$ in target sequence PolyC-2-NGA. By contrast, VQR-Cas9 BEs harboring CDA1 truncations had a much narrower activity window and predominantly edited positions $C_{-17}$ and $C_{-18}$ in target

sequence PolyC-1-NGA and $C_{-17}$ and $C_{-18}$ in sequence PolyC-2-NGA (Fig. 1c; Supplementary Fig. 1). Interestingly, the largest truncation, nCDA1Δ188-VQRBE3, even discriminated to some extent between the two positions in that $C_{-18}$ was edited nearly twice as efficiently as $C_{-17}$ in sequence PolyC-1-NGA (Fig. 1c).

**Fig. 1 High-precision base editing at target sites containing non-NGG PAMs. a** Structure of nCDA1-BE3 in comparison to base editors harboring CDA1 truncations (ΔCDA1). nSpCas9: *Streptococcus pyogenes* Cas9 nickase; XTEN: synthetic linker sequence;[15] UGI: uracil DNA glycosylase inhibitor; NLS: nuclear localization signal. **b** Cas9 variants with altered PAM specificities. **c–g** BE variants with CDA1 truncations mediate high-precision base editing at target sites comprised of multiple cytidines (polyC targets; for sequences see Supplementary Table 1). The *x*-axis shows the Cs in the target sequence with their position relative to the PAM indicated (Supplementary Table 1). The *y*-axis (C-to-T editing in %) represents the percentage of total sequencing reads with the target C converted to T. **c** Analysis of base editing precision of VQR-Cas9 BEs fused to selected C-terminally truncated versions of CDA1 (for the complete deletion series, see Supplementary Fig. 1). For comparison, the BE carrying the full-length CDA1 and the nCDA1-BE3 editor are also included. **d** Analysis of base editing precision of VRER-Cas9 BEs fused to C-terminally truncated CDA1 versions (for the complete deletion series, see Supplementary Fig. 2). **e** Analysis of base editing precision of xCas9 BEs fused to C-terminally truncated CDA1 versions (for the complete deletion series, see Supplementary Fig. 3). **f, g** Analysis of base editing precision of SpCas9-NG BEs fused to C-terminally truncated CDA1 versions (for the complete deletion series, see Supplementary Fig. 4). Values and error bars represent the mean and standard deviation of three independent biological replicates. Source data underlying **c–g** are provided as a Source Data file.

The VRER-Cas9 variant recognizes the PAM sequence NGCG (Fig. 1b). For unknown reasons, the full-length VRER-Cas9 BE (nCDA1-VRERBE3) showed almost no detectable activity on target sequence PolyC-3-NGCG, while it edited target sequence PolyC-4-NGCG with high efficiency but low precision (Fig. 1d; Supplementary Fig. 2). By contrast, our truncated variants efficiently edited both target sequences and displayed greatly superior editing precision on sequence PolyC-4-NGCG (Fig. 1d; Supplementary Fig. 2).

Recently, two Cas9 variants, designated xCas9 and SpCas9-NG, were developed that show greatly relaxed PAM recognition specificity and, instead of NGG, recognize the minimal PAM sequence NG[26,27]. When tested on three non-NGG target sites (PolyC-1-NGA, PolyC-5-NGC, and PolyC-6-NGT), xCas9-derived BEs displayed detectable activity only on one of the three sites (PolyC-5-NGC; Fig. 1e; Supplementary Fig. 3), consistent with several recent studies that reported low genome editing activity and strong sequence context dependence of xCas9 (e.g., refs. [22,26,29–31]). Within the single target sequence that was recognized by xCas9-derived BEs, the full-length xCas9 BE (nCDA1-xBE3) recognized all five Cs in the editing window with similarly low efficiency (of approximately 10%; Fig. 1e). By contrast, the best-performing truncated variant, nCDA1Δ194-xBE3, edited position $C_{-18}$ with high selectivity and strongly enhanced efficiency (of more than 35%; Fig. 1e).

BEs constructed with SpCas9-NG edited all three non-NGG target sites (Fig. 1f, g; Supplementary Fig. 4). Compared to the full-length BE (nCDA1-NGBE3), the truncated versions again exhibited superior editing preference. While the full-length BE edited 4–6 nucleotides with comparable efficiency, the truncated versions predominantly edited one or two nucleotides (Fig. 1f, g; Supplementary Fig. 4). Typically, position $C_{-18}$ was most efficiently recognized, but dependent on the target site, some BEs also edited $C_{-17}$ (e.g., nCDA1Δ194-NGBE3 in PolyC-1-NGA) or $C_{-19}$ (e.g., nCDA1Δ194-NGBE3 in PolyC-6-NGT; Fig. 1g) at high efficiency. For comparison, we also tested the reciprocal fusions harboring the SpCas9 variants at the N-terminus (cCDA1-VQRBE3, cCDA1-VRERBE3, cCDA1-xBE3, and cCDA1-NGBE3). These fusions showed a narrower activity window than the C-terminal fusions, but did not reach the specificity of the best-performing fusions with truncated CDA1 versions (Supplementary Figs. 1–6). When target sites upstream of the wild-type PAM of Cas9, NGG, were tested, the SpCas9-NG-derived BEs displayed reduced editing activity compared to wild-type Cas9-derived BEs (Supplementary Figs. 7 and 8). This finding is consistent with recent studies that reported lower genome editing activity of SpCas9-NG on canonical NGG PAMs[26,32].

Taken together, our findings indicate that BEs with truncated CDA1 sequences tolerate replacement of Cas9 with variants that recognize alternative PAMs, including PAMs with greatly relaxed specificity such as NG. The high efficiency and accuracy of these editors greatly expand the editing scope of high-precision BEs.

**Engineering of A3A-based precision BEs.** In an attempt to develop additional high-precision BEs that selectively edit nucleotide positions other than $C_{-18}$, we generated fusions of several deaminases to nCas9 by omitting a linker sequence between the two proteins. This approach was taken to investigate the possibility that these deaminases inherently harbor a linker-like fragment at their C-terminus, as discovered recently for CDA1[23]. Omission of the synthetic linker sequence and removal of C-terminal amino acids was shown to greatly increase the base editing accuracy of CDA1-derived BEs while retaining high editing efficiency[23].

Six different deaminases were tested by fusing nCas9 directly to their C-terminus (Supplementary Fig. 9a). The fusion proteins were then assayed for their base editing efficiency on two polyC-containing target sites (Supplementary Fig. 9b). The BE based on the human cytidine deaminase APOBEC3A (A3A; ref. [22]), referred to as hA3A-NL-BE3, displayed the best performance in that it conferred the highest editing efficiency on both target sequences, but also showed the broadest editing window (Supplementary Fig. 9). We, therefore, chose A3A for further optimization.

For comparison, we also generated an A3A-BE3 editor with the standard XTEN linker[18]. Surprisingly, we observed that hA3A-NL-BE3 (for brevity subsequently referred to as A3A-NL-BE3) showed a slightly broader editing window than A3A-BE3 and also caused a shift in the most strongly edited (central) positions (Supplementary Fig. 10; Supplementary Table 2), despite the shorter connection between the cytidine deaminase domain (A3A) and the nCas9 domain of the fusion protein. This may be attributable to linker removal slightly altering the spatial structure of the fusion protein (and, in this way, affecting positioning of the deaminase domain on the target sequence), and would be consistent with the variable effects of linker engineering seen in previous studies[23,33]. The editing efficiency of both BEs was similar at both tested sites (Supplementary Fig. 10), possibly suggesting that the C-terminus of A3A is extraordinarily flexible.

A3A-based BEs were reported to exhibit a lower dependence on the sequence context, reduced sensitivity to DNA methylation and a wider editing window[11,22,34]. To test if the precision of these BEs can be improved by narrowing the activity window, we constructed a series of truncations at the C-terminus of A3A and determined their impact on base editing (Fig. 2a). Previously, we showed that the major gain in site selectivity for CDA1-based BEs was seen with the removal of at least 13 amino acids from the C-terminus (nCDA1Δ195-BE3; ref. [23]). Alignment of A3A with CDA1 revealed that the 13 amino acid CDA1 truncation corresponds to residue 194 of A3A (Supplementary Fig. 11a).

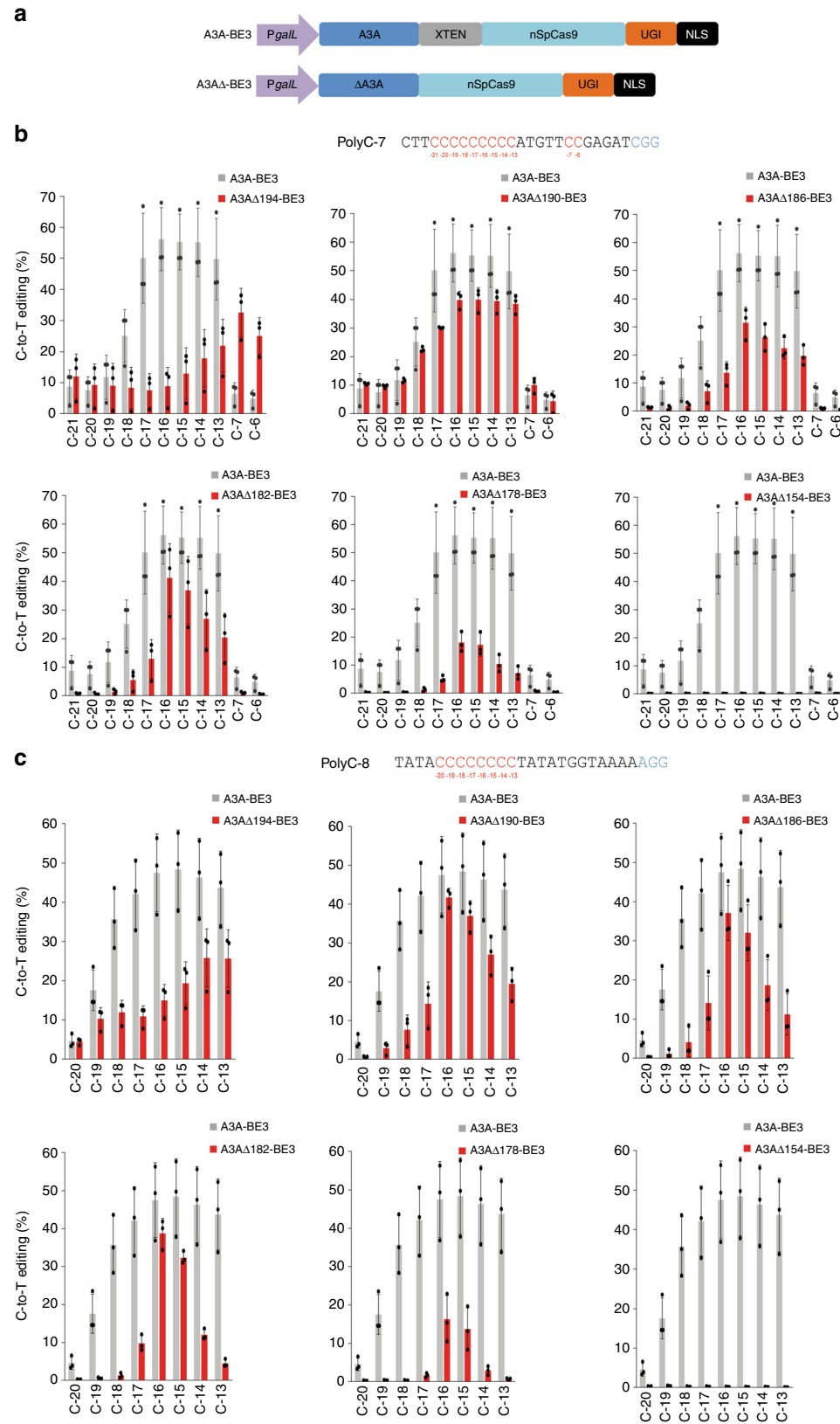

We generated six BEs with C-terminally truncated A3A versions fused to nCas9 (Supplementary Fig. 11b) and tested them on two polycytidine motifs (Fig. 2b). While deletion of 5 or 9 amino acids (A3AΔ194-BE3 or A3AΔ190-BE3) did not narrow the editing window within target sequence polyC-7 compared to A3A-BE3, deletion of 17 amino acids (A3AΔ182-BE3) made the editing

significantly more specific in that A3AΔ182-BE3 preferentially edits position $C_{-15}$ or $C_{-16}$ (Fig. 2b; Supplementary Table 2). When tested on target sequence polyC-8, the truncated editors A3AΔ190-BE3, A3AΔ186-BE3 and A3AΔ182-BE3 displayed improved specificity. For example, A3AΔ182-BE3 exhibits a strong preference for positions $C_{-15}$ and $C_{-16}$, while showing

**Fig. 2 Base editors with C-terminally truncated A3A sequences exhibit narrowed editing windows. a** Structure of A3A-BE3 and BEs with A3A truncations (A3AΔ-BE3 variants). The various A3A truncations tested are shown in Supplementary Fig. 11. **b**, **c** Effects of C terminal truncations of the A3A domain on the width of the editing window of A3AΔ-BE3s. All base editor variants were tested on both the polyC-7 (**b**) and polyC-8 (**c**) sites (see "Methods" section). Cs within each target region are indicated in red, with the number below indicating their distance from the PAM (blue). The C-to-T conversion efficiencies are plotted for all Cs within the protospacer, and shown in comparison to the A3A-BE3 base editor with the full-length A3A (gray bars). Values and error bars represent the mean and standard deviation of three biological replicates. Source data underlying **b**, **c** are provided as a Source Data file.

greatly reduced editing activity at the neighboring positions $C_{-17}$ and $C_{-14}$ (Fig. 2c; Supplementary Table 2). When more than 17 amino acids were removed from the C-terminus of A3A, editing efficiency was strongly reduced (A3AΔ178-BE3) and eventually lost completely (A3AΔ154-BE3; Fig. 2b, c).

To confirm the superior precision of the truncated editors A3AΔ190-BE3, A3AΔ186-BE3, and A3AΔ182-BE3, we compared the base editing outcomes when targeting different cytidines within the yeast Can1 gene[23]. Each of the five tested sites contains one or two target Cs in different distances from the PAM, ranging from position $C_{-19}$ to position $C_{-11}$ (Fig. 3a). Canavanine-resistant colonies clones can arise only when C-to-T base editing occurs and results in synthesis of an inactive gene product[23]. While the BE with the full-length A3A (A3A-BE3) non-selectively edited all Cs within a window of nine nucleotides (Fig. 3b), the BEs containing truncated A3A versions mainly edited positions $C_{-15}$ or $C_{-16}$, confirming the results obtained with polycytidine target sequences (Fig. 2b).

It was recently reported that mutations in A3A (N57G mutation in an A3A variant dubbed eA3A) can reduce bystander editing frequency by enhancing the preference of the editor for TCR motif (with R being A or G; ref. [22]). We, therefore, generated an eA3A-BE3 editor and compared it with our best-performing truncated A3A BEs. We found that eA3A, although mainly editing $C_{-15}$ or $C_{-16}$, suffered from reduced editing activity (Fig. 3b), suggesting relatively poor editing at non-TCR sites.

It has been reported that A3A-derived BEs can induce significant transcriptome-wide off-target editing at the RNA level. Specific amino acid substitutions (R128A or Y130F) in A3A largely eliminate these off-target activities[35,36]. We, therefore, investigated the effect of each of these two mutations on the width of the base editing window and the BE activity when combined with proper A3A truncations. Introduction of either of the two mutations into A3A-BE3 neither reduced the base editing efficiency, consistent with previous findings[35], nor did it affect the base editing window (Supplementary Fig. 12). When we combined these mutations with the two optimal A3A truncations (A3AΔ186 and A3AΔ182), we found that Y130F, but not R128A, in combination with the A3A version truncated at residue 186 (i.e., BE variant A3A(Y130F)Δ186-BE3) displays a base editing window and an editing efficiency similar to A3AΔ186-BE3 (Supplementary Fig. 12), and thus should be used to suppress off-target RNA editing.

Together, these data demonstrate that the A3A deaminase can be engineered to obtain high-precision BEs that predominantly edit position $C_{-15}$ or $C_{-16}$, while retaining high editing efficiency.

**Analysis of genome-wide off-target editing**. Recently, cytosine BEs were reported to produce substantial genome-wide off-target effects that are largely independent of the sgRNA[37,38]. Since a narrower editing window means fewer target nucleotides, we envisioned that our narrow-window BEs could also reduce the off-target DNA editing. We, therefore, investigated off-target editing in yeast cells treated with nCDA1-BE3, cCDA1-BE3, nCDA1Δ190-BE3, and a no BE control, in combination with an sgRNA targeting a Can1 site (Supplementary Fig. 13a).

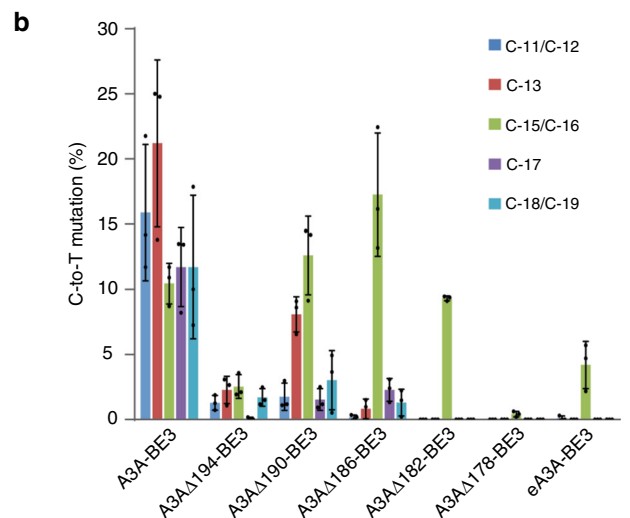

**Fig. 3 Base editing outcomes of A3A-BE3, truncated A3AΔ-BE3 variants and the recently optimized editor eA3A-BE3 (ref. [22]) when targeting specific sites in the yeast Can1 gene. a** Sequences of the five target sites (containing Cs at different positions). Target Cs are indicated in red and numbered relative to the PAM (blue). Edited clones were identified by using the canavanine selection strategy (see "Methods" section). **b** Base editing efficiency and precision. The x-axis represents the target Cs within the protospacers. The y-axis shows their C-to-T mutation frequency (see "Methods" section). Values and error bars represent the mean and standard deviation of three independent biological replicates. Source data underlying **b** are provided as a Source Data file.

Canavanine selection was used to isolate colonies harboring on-target editing events. The truncated CDA1 version Δ190 was chosen for this experiment, because we had previously shown that this version displays high editing precision as well as high editing efficiency for most tested sites[23]. For all constructs, cultures grown from three different transformed colonies were mixed, followed by genomic DNA isolation and whole-genome sequencing (Supplementary Fig. 13b). Expectedly, the three BE variants showed comparable numbers of indels as the no BE control (Fig. 4a). When the total number of SNVs (single-nucleotide variants) was analyzed, the full-length fusions were found to display many more SNVs than the control, in agreement with the previous reports on off-target effects of cytosine BEs[37,38]. However, the truncated version exhibited a substantially reduced number of SNVs that was only slightly higher than that of the

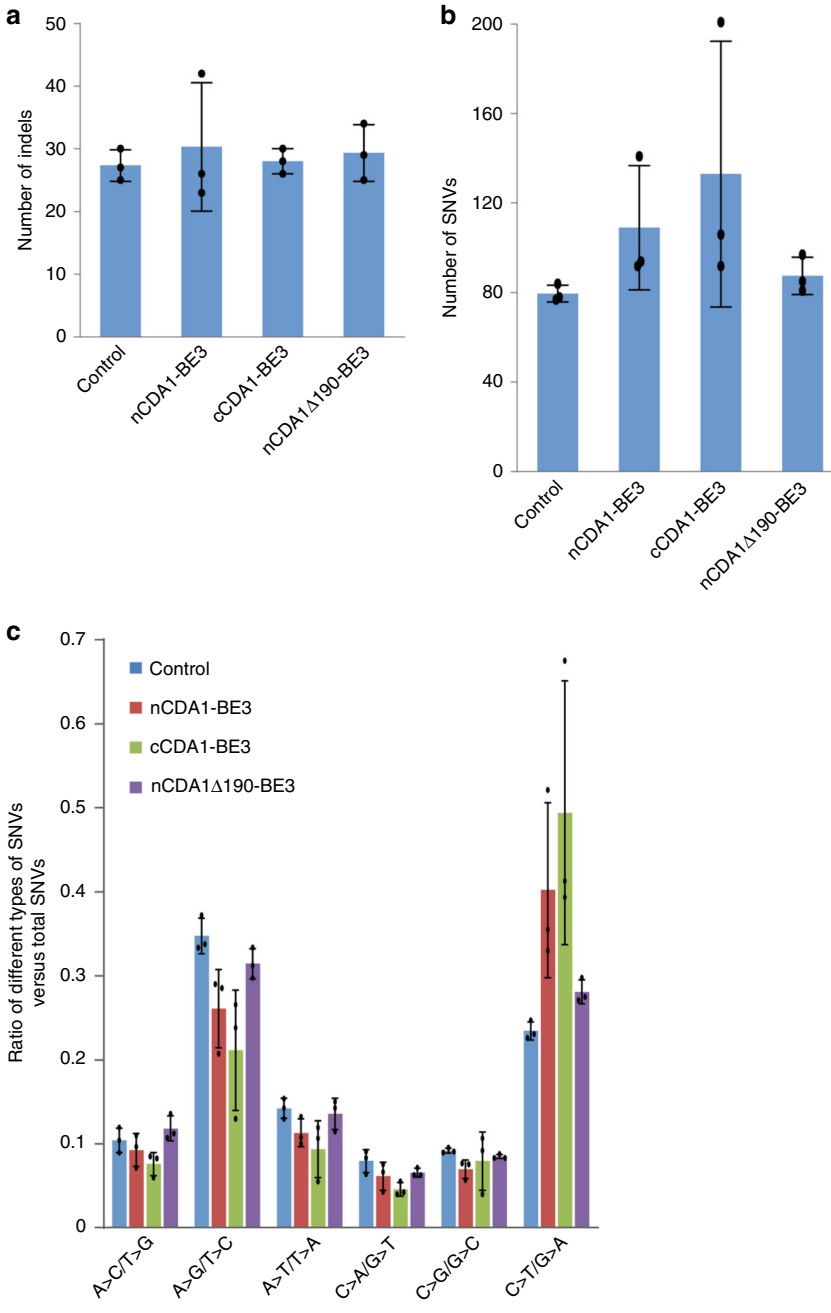

**Fig. 4 Analysis of off-target editing. Genetic changes that occurred in strains harboring nCDA1-BE3, cCDA1-BE3, nCDA1Δ190-BE3 or a control plasmid without a BE construct were identified by whole-genome sequencing. a, b** Comparison of the total number of detected indels (**a**) and SNVs (**b**). For information on the constructs used and the experimental workflow, see Supplementary Fig. 13. **c** The mutation frequency of different types of SNVs in cells treated by the three base editors and the control. The sgRNA was designed to target site *Can1-4* (Supplementary Table 1). Values and error bars represent the mean and standard deviation of three independent biological replicates. Source data are provided as a Source Data file.

negative control (Fig. 4b). We also analyzed the mutation types and found that, in nCDA1-BE3 and cCDA1-BE3, the frequency of C-to-T (G-to-A) transitions was significantly higher than in the control and the truncated base editor nCDA1Δ190-BE3 (Fig. 4c). These findings indicate that high editing precision of BEs can contribute to reduced non-specific editing at off-target sites.

**Guidelines for the choice of the optimal cytidine BE.** Three different cytidine deaminases (APOBEC1, CDA1 and APO-BEC3A) have been engineered to produce efficient cytosine BEs,

modify PAM specificities, and alter position and width of the editing window (e.g., refs. [18,23]; this work). BE variants with different properties have been obtained that differ in their suitability for (i) different target sequences and (ii) different positions of the C to be edited within the protospacer.

The results obtained with the various BE variants and target sequences tested are complex, and no simple rules can be deduced. However, there is now sufficient information available to define some guidelines for the choice of the best BE depending on the position of the C, the sequence context and the presence or absence of bystander Cs (Table 1). For example, if the target C is located at position $C_{-19}$ relative to the PAM and no bystander C

**Table 1 Recommendations for BE selection for precision cytosine base editing.**

| Distance of target C from PAM | Bystander | Recommended BEs |
|---|---|---|
| <−19 | no | nCDA1-BE3, nCDA1Δ198-BE3, A3A-NL-BE3 |
| −19 | no | nCDA1-BE3, nCDA1Δ198-BE3, A3A-NL-BE3 |
|  | CCDDD | cCDA1-BE3 |
| −18 | no | nCDA1Δ198-BE3, n/cCDA1-BE3, A3A-NL-BE3 |
|  | NCN | nCDA1Δ(194-188)-BE3 |
| −17 | no | nCDA1Δ198-BE3, cCDA1-BE3, A3A-BE3, nCDA1-BE3, BE3 |
|  | DCN | nCDA1Δ(194-188)-BE3 |
|  | CCDDDD | BE3 |
| −16 | no | BE3, nCDA1Δ198-BE3, A3A-BE3 |
|  | DDDCC | cCDA1-BE3 |
|  | TCC | cCDA1-BE3, YEE-BE3, BE-PAPAPAP |
|  | NCD | A3AΔ182-BE3, A3A(Y130F)Δ186-BE3 |
| −15 | no | BE3, nCDA1Δ198-BE3, A3A-BE3 |
|  | CDCD | BE-PAPAPAP, A3AΔ182-BE3, A3A(Y130F)Δ186-BE3, YEE-BE3 |
|  | DCN | A3AΔ182-BE3, A3A(Y130F)Δ186-BE3 |
|  | RCCD | BE-PAPAPAP |
| −14 | no | BE3, A3A-BE3, nCDA1-BE3 |
|  | DDCC | BE-PAPAPAP |
| >−14 | no | A3A-BE3, nCDA1-BE3, BE3 |

The C to be edited is underlined, "no bystander" means absence of other Cs from the activity window of the BEs. N: any nucleotide (including a possible bystander C), D: not C (i.e., A, G or T), R: A or G.

is present, three BEs can be recommended: nCDA1-BE3, nCDA1Δ198-BE3 and A3A-NL-BE3. If the target C is in the same position ($C_{−19}$), but has a bystander C directly upstream (CCDDD motif, with D being any nucleotide but C), cCDA1-BE3 would be the best choice[23]. If the target C is located at $C_{−18}$ and has a bystander C in its vicinity (NCN motif, with N being any nucleotide, including a possible bystander C), BEs with C-terminal truncations of CDA1 (Δ194 to Δ188) are recommended (Fig. 1; ref. [23]), and it may be advisable to test two or three different truncations. For editing at $C_{−16}$ with a 5′ bystander C (NCD context) or editing at $C_{−15}$ with a 3′ bystander C (DCN), A3AΔ182-BE3 and A3A(Y130F)Δ186-BE3 are the editors of choice (Figs. 2, 3; Table 1).

With our set of narrow-window BEs, many more disease-causing T-to-C (or A-to-G) mutations now can, in principle, be corrected in a precise manner (Supplementary Fig. 14). For example, a T-to-C mutation at position 497 of the coding region of the human gene encoding presenilin-1 (*PSEN1*-L166P mutation) is associated with early-onset Alzheimer's disease[39]. This mutation can be corrected by a BE that has this C within its predicted editing window at position −18 relative to the PAM sequence NG (for a suitable sgRNA sequence, see Supplementary Table 1). Precision is important here, because an additional C is present immediately adjacent to the target C (at position 496), which also lies within the editing window (−19 relative to the PAM). Using precision BEs with CDA1 truncations, this C now can be targeted much more accurately (Table 1). Similarly, an A-to-G mutation at position 980 of the coding region of the tyrosinase-encoding gene (representing a T-to-C mutation in the complementary strand) causes oculocutaneous albinism (*TYR*-Y327C mutation[8]). The target C is in a TCAC motif and located in position −15 of the PAM sequence AGG. Therefore, this mutation can be precisely corrected with the BEs A3AΔ182-BE3 or A3A(Y130F)Δ186-BE3 (Supplementary Table 1; Table 1).

It is important to note that there are still cases in which the currently available BEs cannot discriminate well between target C and bystander C. One such case is $YCC_{−15}$ (Y being T or C), due to the editing preference of $C_{−16} ≥ C_{−15}$ displayed by our A3A-based BEs (Fig. 2). Additional BEs will need to be developed to fill these gaps. However, the currently available BEs, in combination with the different PAM sequences that can be exploited (Fig. 1),

cover already a large sequence space, so that it will be possible to select an efficient BE for most target sequences (Table 1).

**Discussion**

In the course of this work, we have (i) extended the applicability of our previously developed high-precision BEs[23] by efficiently targeting non-NGG PAM sequences, and (ii) generated BEs that target alternative positions in the protospacer with high accuracy. Together, the BE variants now available cover widely different sequence contexts, PAM sequences and distances of the editing site from the PAM. Thus, it is now possible to select a suitable BE from the available set (Table 1) for nearly any sequence motif.

Two major factors are known to affect the editing precision of BEs: (i) deaminase activity, including substrate binding affinity and catalysis, and (ii) conformation of the complex of Cas9, deaminase, sgRNA, and ssDNA. Specific mutations in the deaminase domain can increase base editing precision[22,33]. Similarly, the use of different deaminases fused to Cas9 and/or the engineering of the spacer region that separates the deaminase domain from Cas9 can alter editing precision. In particular, shortening the linker that connects Cas9 and the deaminase (e.g., by removing nonessential fragments from the termini of the deaminase[23]) can reduce the enzyme's activity towards cytosine that are not optimally presented to the deaminase active site, thus increasing editing precision. Importantly, different from previous approaches[22,33], the strategies we applied to narrow the editing window of BEs do not entail reduction of the deaminase activity and thus, combine superior editing precision with high editing efficiency.

It should be noted that the poly(C) motifs used to assay our BEs for specificity represent the worst-case scenario, in that such long homopolymeric cytidine stretches would only rarely be useful in vivo targets of base editing. Consequently, in most cases, a BE can be selected that likely operates with high precision. For example, if there is no bystander C immediately 5′ and/or 3′ of the target C, our A3AΔ182-BE3 edits the target nucleotide highly specifically, without any detectable off-site background (Fig. 3b).

BEs were recently reported to cause substantial off-target effects in both genomic DNA[37,38] and RNA[35,36]. Using whole-genome sequencing, we found that the precision BE with a

truncated CDA1 exhibited reduced off-target DNA editing in comparison to BEs containing the full-length deaminase (Fig. 4). The likely explanation for this finding is that a narrower editing window means fewer target sites, also including off-target sites. Thus, our findings suggest that the design of BEs with improved editing precision also contributes to addressing potential problems caused by off-target effects. Although we have not tested off-target RNA editing by CDA1- and A3A-derived BEs directly, we established that precision-increasing truncations can be combined with specific point mutations in the deaminase which are known to eliminate off-target RNA editing (A3A(Y130F) Δ186-BE3), while maintaining DNA editing precision and efficiency (Supplementary Fig. 12). It is also noteworthy in this regard that, compared to APOBEC1 and A3A, CDA1-based BEs were reported to induce much less editing of RNA[40].

In this work, we used baker's yeast to improve the precision of BEs through protein engineering. Although we have not yet applied our BEs to base editing in animals and plants, it is well established that, like most genome editing tools, BEs are readily transferrable between organisms and retain their properties[14,19,41]. This is likely due to the simple molecular structure of CRISPR-Cas systems[42] and the absence of species-specific factors involved in the editing reaction and/or target site recognition. It will be interesting to apply similar strategies to adenine BEs to improve their site selectivity and generate a set of adenine BEs from which the best option can be chosen depending on the location of the target nucleotide and the surrounding sequence context, similar to the cytosine BE set listed in Table 1.

High-precision BEs represent essential tools for future applications of DNA editing, especially in gene therapy and precision breeding[10,20,21]. For example, the correction of mutations causing hereditary diseases in humans requires high accuracy of the BEs employed in that introduction of undesired mutations in the vicinity of the target nucleotide must be avoided[15,20]. Although the efficiency of currently available BEs may not yet be high enough for many applications in in vivo gene therapy, the BEs will be useful in ex situ therapeutic applications and will also find applications in other fields such as crop breeding (given that much of the useful genetic variation in plant breeding represent point mutations).

Finally, while our narrow-window BEs are sufficiently accurate for many target sequences (Fig. 3b), they are not yet perfect in sequence contexts that require absolute discrimination between neighboring cytidines (Figs. 1, 2). Thus, further improvements will be needed to eliminate bystander editing even in homopolymeric cytidine tracts.

## Methods

**Yeast strains and growth conditions.** *Saccharomyces cerevisiae* BY4743 (diploid, *MAT* a/α, *his3Δ1/his3Δ1, leu2Δ0/leu2Δ0, LYS2/lys2Δ0, met15Δ0/MET15, ura3Δ0/ura3Δ*) was used as host strain for genome editing experiments. Cells were grown under non-selective conditions in liquid YPAD medium (2% Bacto peptone, 1% Bacto yeast extract, 2% glucose, 0.003% adenine hemisulfate). For culture in Petri dishes, the YPAD medium was solidified with 2% agar. Selection of transgenic yeast clones was conducted on synthetic complete (SC) medium (6.7 g/L of Difco Yeast Nitrogen Base, 20 g/L glucose) with a mixture of appropriate amino acids (deficient in uracil and leucine; SC-U-L) based on the use of the *URA3* and *LEU2* markers. Yeast cultures in liquid medium were grown at 28 °C on a rotary shaker at 185 rpm.

**Cloning and polymerase chain reaction (PCR).** PCR was performed with Phusion High-Fidelity DNA Polymerase (ThermoFisher) following the manufacturer's instructions. All primers used in this study are listed in Supplementary Table 3. Cloning of vectors for yeast transformation and plasmid amplification were carried out in the *Escherichia coli* laboratory strain DH5α. Vectors containing the *Streptococcus pyogenes cas9* gene (p415-GalL-Cas9-CYC1t) and a chimeric guide RNA construct (p426-SNR52p-gRNA.CAN1.Y-SUP4t) were obtained from Addgene (Cambridge, MA, USA).

To generate CDA1-BE3 variants with VQR-Cas9, the three required point mutations (D1135V/R1335Q/T1337R) were introduced into *cas9* by PCR with

primers harboring the desired mutations, and the resulting three PCR products were cloned into the NruI/NcoI-digested BE3 to obtain VQR-BE3 with the help of the In-Fusion HD Cloning Kit (Clontech, Mountain View, CA, USA). The mutated fragment was then released by digesting VQR-BE3 with NruI and MIuI, followed by ligation into the similarly digested CDA1 BE plasmid[23]. To construct VRER-BE3 variants, three fragments containing the four mutations (D1135V/G1218R/R1335E/T1337R) were PCR-amplified followed by cloning into the NruI/MIuI-digested VQR-BE3. The mutated fragment was then excised by digesting VRER-BE3 with NruI and MIuI, and ligated into the CDA1 BE construct cut with the same enzyme combination. For the generation of SpCas9-NG BE3 variants, four fragments containing the seven mutations (R1335V/L1111R/D1135V/G1218R/E1219F/A1322R/T1337R) were PCR-amplified followed by cloning into the NruI/MIuI-digested vector VQR-BE3. The mutated fragment was released by digesting SpCas9-NG-BE3 with NruI and MIuI and cloned into the similarly cut CDA1 BE plasmid. For the construction of xCas9 variants, plasmid xCas9 (3.7)-BE3 (obtained from Addgene) was digested with the restriction enzymes Sbf1 and AscI. The resulting 3.7 kb fragment was then inserted into the CDA1 BE construct digested with Sbf1 and AscI. To obtain cCDA1-BE3 variants, the mutated fragments were PCR-amplified using the corresponding BE3 variant as template and cloned into the NurI/SphI-digested cCDA1-BE3 plasmid[23].

To generate hA3A, hA3B, hA3G, hAID, mAID, cAICDA and truncated hA3A BEs, the deaminase genes were PCR-amplified from plasmid clones (provided by the laboratory of Dr. Jia Chen, Shanghai, China, and obtained from Addgene) together with part of the *cas9* sequence, and then ligated into the SpeI-digested BE3 vector. To produce A3A(R128A)-BE3, A3A(Y130F)-BE3 as well as eA3A-BE3, the point mutations (R128A, Y130F, and N57G) were introduced into A3A with primers containing the appropriate mutations.

For the construction of plasmids expressing sgRNAs that target specific sites (Supplementary Table 1), the protospacer sequences were introduced by PCR amplification (as part of the primer sequence; see Supplementary Table 3), and the resulting fragments were cloned into the ClaI/KpnI-digested vector p426-SNR52p-gRNA.CAN1.Y-SUP4t with the In-Fusion HD Cloning Kit (Clontech).

**Transformation, DNA isolation, and sequencing of PCR products.** Yeast cells were transformed with the LiAc/SS carrier DNA/PEG method using 0.5–1 µg for each plasmid DNA (ref. [23]). Yeast genomic DNA was extracted using a lithium acetate-SDS-based protocol[23]. PCR products were purified with the PCR Purification kit (Macherey-Nagel, Düren, Germany) and sequenced using the dideoxy chain termination method.

**High-throughput DNA sequencing and data analysis.** For high-throughput sequencing, yeast colonies were suspended in 3 mL SC-L-U medium with 2% glucose and grown to stationary phase. 0.8 mL samples of each culture were then pelleted, washed twice with sterile water, and resuspended in SC-L-U induction medium supplemented with 2% galactose and 1% raffinose to an $OD_{600} \approx 0.3$. Subsequently, the cells were grown for 20 h at 28 °C on a rotary shaker at 185 rpm. Samples of 0.5 mL of each culture were submitted to genomic DNA extraction and the regions targeted by base editing were amplified by PCR with index tag-containing primer pairs for multiplexing (Supplementary Table 3). PCR amplification was conducted with the Phusion High-Fidelity DNA Polymerase (ThermoFisher, Waltham, MA, USA) according to the manufacturer's instructions, followed by purification of the amplified fragments with the NucleoSpin Gel and PCR clean-up kit (Machery-Nagel). The resulting index-labeled PCR products were then pooled at equal molar ratios. PCR-free library construction, high-throughput sequencing, demultiplexing by assigning reads to samples, and data processing (including removal of adaptor sequences, contaminations, and low-quality reads) were performed commercially (BGI, Hong Kong). DNA sequencing was carried out on an Illumina HiSeq 4000 platform in a paired-end way to obtain 150 nt read length for each side. On average, more than 100,000 reads were obtained for each sample. The clean FASTQ files obtained after data filtering were further analyzed with Python scripts[23] (available at https://github.com/zfcarpe/Cas9Sequencing).

Bioinformatic analysis of the ClinVar database[8] for human disease-associated mutations was performed with Python scripts (available at https://github.com/zfcarpe/Cas9Sequencing), which were adapted from a previous report[33]. Modifications included the incorporation of additional BEs, distinct priority ranking of target C positions (C$_{-18}$ > C$_{-17}$ > C$_{-19}$) and PAM identification for BE variants described in this work. Sequence alignments were created by CLUSTAL W[43] and graphically formatted with the help of the ESPript 3.0 server[44].

**Can1 mutagenesis.** Yeast cultures were grown from single colonies (suspended in 3 mL SC-L-U medium with 2% glucose) to stationary phase. Samples of 0.8 mL were then pelleted, washed twice with sterile water (to remove residual glucose), and resuspended in SC-L-U induction medium supplemented with 2% galactose and 1% raffinose to an $OD_{600}$ of ~0.3. The cells were then incubated under shaking at 185 rpm for 20 h, followed by plating on YPAD or SC-Arg medium supplemented with 60 µg/mL L-canavanine (Merck, Darmstadt, Germany), and the colony number per plate was determined after incubation for 3 days. The frequency of induced C-to-T mutations in *Can1* was determined as the ratio of the colony count on canavanine-supplemented plates to the colony count on drug-free plates. Each

experiment was performed at least in triplicate on different days. Control cultures (not expressing BEs) did not yield canavanine-resistant colonies.

**Whole-genome sequencing for analysis of off-target editing**. Yeast strains expressing nCDA1-BE3, cCDA1-BE3 or nCDA1Δ190-BE3 (or harboring a no-BE control plasmid) and the sgRNA targeting the *Can1-4* site were incubated in SC-L-U medium with 2% glucose followed by transfer to induction medium supplemented with 2% galactose and 1% raffinose for 20 h. After induction, appropriate volumes of each culture were plated on YPAD or SC-Arg medium with 60 μg/mL L-canavanine (Merck, Darmstadt, Germany), and grown for 3 days to allow for formation of resistant colonies. Three colonies from each plate were randomly picked, suspended in YPAD medium and incubated overnight. Equal volumes from the three cultures were then mixed and genomic DNA was extracted using the Wizard Genomic DNA purification Kit (Promega, WI, USA) following the manufacturer's instructions. Test of DNA sample quality, library construction, high-throughput sequencing and bioinformatics analysis of the raw data were performed commercially (BGI, Hong Kong). DNA sequencing was carried out on an Illumina HiSeq X Ten platform.

**Reporting summary**. Further information on research design is available in the Nature Research Reporting Summary linked to this article.

## Data availability
The data supporting the findings of this study are available within the paper and its supplementary information files. High-throughput sequencing data have been deposited in the National Center for Biotechnology Information Sequence Read Archive database under accession code PRJNA562458. The source data underlying Figs. 1–4 and Supplementary Figs 5, 6, 8–10 and 12 are provided as a Source Data file.

## Code availability
Python scripts used in this study are available at GitHub (https://github.com/zfcarpe/Cas9Sequencing).

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

## Acknowledgements
We thank Dr. Youjun Zhang (MPI-MP) for providing the yeast strain. This research was supported by the Max Planck Society and a grant from the European Union (Horizon 2020, Newcotiana, 760331-2) to R.B.

## Author contributions
R.B., D.K. and J.T. designed the research. J.T. performed most of experiments. F.Z. performed bioinformatic analyses. R.B. and J.T. wrote the paper.

## Competing interests
A patent application on high-precision base editors has been filed with R.B., J.T. and D.K. as inventors (PCT/EP2019/056335).
