## [Peer Review File · Nature Communications]

Reviewers' Comments:

Reviewer #1:

Remarks to the Author:

To authors:

Tan et al. Expanding the genome-targeting scope and the site selectivity of high-precision base editors.

This manuscript by Tan and colleagues presented the development of new high-precision BEs that expand the versatility of base editing in yeast. The authors mainly replaced the Sp-nCas9 sequence with the reported nCas9 variants which recognize different non-NGG PAMs. Meanwhile, they tested the six different deaminases by fusing nCas9 directly to their C-terminus and found A3A displayed the best performance. Lastly the authors made the guidelines for the choice of the optimal cytidine BE. In general, this work provided information for base editing, but it is not clear that this advance on its own meets the novelty standards for this journal.

Specific points,

1. Although the authors provided new base editors for the high-precision, all the nCas9 and deaminases have been reported in the previous research.
2. The authors tested few target sites, for example, only one target site for nCDA1- Δ 195-VQRBE3 and nCDA1- Δ 194-VQRBE3. It is not enough to get the solid conclusions.
3. The safety about base editing causes great attentions because of the off-target. The authors did not analyze the off-target for the new base editors.
4. BE3 is not fit for RCC-15D if R was A in Table 1.
5. More information about the influence factors on editing precision of BEs should be provided.

Reviewer #2:

Remarks to the Author:

The manuscript authored by Tan et al reports the development of new cytosine base editors with diversified high-precision editing scopes and expanded PAMs in yeast. The authors first fused CDA1 truncations with nCas9 variants and obtained diverse cytosine base editors editing at C-18 precisely with expanded PAMs in addition to NGG. Additionally, the authors also tested the editing signatures of various APOBEC3A truncations fusing to nCas9, and obtained cytosine base editors editing at C-15/C-16 precisely. Generally, this manuscript provide new tools for C-T conversions with diversified editing scopes and expanded versatility, which are useful for base editing applications. I have the following points to improve the manuscript:

1. The authors observed that CDA1 fusing to the C-terminus of SpCas9-NG(cCDA1-NGBE3) narrowed the editing window. However, they did not evaluate the editing signatures of CDA1 fusing to the C-terminus of other nCas9 variants including VQR-nCas9, VRER-nCas9 and nxCas9. Additionally, the authors should demonstrate whether CDA1 truncations fusing to the C-terminus of nCas9 variants would change the activity or editing window.
2. It has been reported that APOBEC3A induced significant off-target C-U conversions at RNA level, and specific mutations could eliminate those off-target activities at RNA level (e.g. Y130F, R128A)(Zhou et al. Nature 2019;Grunewald et al. NBT 2019). In consideration of the future applicability of newly established APOBEC3A truncations fusing to nCas9, it is important to examine whether APOBEC3A truncations still have off-target activities at RNA level, and whether established mutations (e.g. Y130F, R128A) affect the activities and editing windows of APOBEC3A truncations.
3. The authors demonstrated that hA3A-NL-BE3 had a broader editing window than A3A-BE3, which indicated the standard XTEN linker might narrow the editing window. However, in Fig.2, the

authors just compare A3A-BE3 (with XTEN linker) with A3Adelta-BE3 (without XTEN linker). The editing signatures of A3A-NL-BE3 and A3Adelta-BE3 with XTEN linker are needed for systematic comparisons.

Reviewer #3:

Remarks to the Author:

In a recent prior paper (Nat. Commun. 10, 439) the authors (Tan et al.) had shown that they could reduce the problem of on-target "bystander mutations" generated by Cas9 base editors by truncating their deaminase domain. This substantially narrowed the window of bases that were efficiently edited, in some cases to just a single preferentially edited base (e.g. base -18 relative to the PAM). In their current paper, the authors seek to extend this work to Cas9 variants developed by others to have altered or relaxed PAM requirements and therefore enhanced targeting range. They fuse six truncated CDA1 domains (AID homologue of sea lamprey) to 4 different Cas9 variants with relaxed PAM requirements to ultimately increase preference for editing position -18 and therefore reduce bystander mutations. All six truncated CDA1 domains have been described previously in the context of the author's prior paper (Nat. Commun. 10, 439) and show a similar editing profile when fused to the SpCas9 variants used in this study.

The authors also screen truncation variants of alternative deaminase domains in an attempt to identify base editors having narrowed editing windows focused on a base different than -18. To accomplish this, they examine C-terminal truncations of cytosine deaminases such as AID, AICDA or APOBEC3A when fused to the native SpCas9. The authors identify three human APOBEC3A truncations with increased editing preference for target positions -15 and -16: $\Delta 190$, $\Delta 186$ and $\Delta 182$, albeit with considerably reduced efficiency (see Figure 2).

This work is of considerable potential interest to those who may wish to use base editing for cellular engineering applications, as it focuses on the two main limitations of this technology (i) an inability to design canonical Sp Cas9 for efficient editing of many potential target bases, due to PAM restrictions, and (ii) the problem of editing additional bases besides the chosen one for any given application. However there are a number of considerations that should be addressed.

Major issues:

Many editing applications, and probably the large majority of therapeutic ones, will require highly efficient base editing (>80%) in relevant cell types, especially when used in multiplexing applications. A key concern of this reviewer is that the data presented in this work provides little support for the ability of the described proteins to enable such high efficiency editing in application-relevant settings. Even though all studies were carried out in yeast – which if anything should provide a less stringent system for gauging performance – levels of targeted editing rarely exceeded 60% with the new constructs. Given the recurrent problem in the field of Cas9 engineering of variant proteins exhibiting insufficient activity in application-relevant settings (e.g. see the authors' own comments on xCas9, and also an examination of high fidelity Cas9 variants in Nat Med 24 1216) it would seem reasonable to require demonstration by the authors that their new editors can perform efficiently at endogenous loci in human cells. Such a demonstration would also serve to show that the narrowed editing window that is crucial to the novelty of this work is preserved in more relevant settings.

This issue seems especially acute in studies of hAPOBEC3A truncations $\Delta 186$ and $\Delta 182$, which rarely exceeded 40% editing in their model target studies, and showed on-target editing efficiency is only ~10-17% at yeast Can1 locus. This makes it difficult to predict if high enough editing will be achievable in application-relevant settings. It also leaves open the question of how bystander mutations will behave if it is possible to increase modification levels by increasing dose.

A second concern is that there is no examination of the cell-wide specificity behavior of the author's truncated deaminase variants. It is unclear how the Cas9-independent off-target profiles of CDA1 or hAPOBEC3A will be impacted by the different truncations on the DNA and RNA level. Establishing that the new variants retain suitable global specificity would seem to be an important requirement given the author's rationale for this work in part as providing improved therapeutic agents.

There is also at times overstatement of the position-selectivity of the editors that have been developed. For example authors say that their truncated deaminases "specifically edit position C18 relative to the PAM sequence" but there is not enough evidence to support such a strong statement. As the authors themselves point out, nCDA1 Δ 188-VQRBE3 can only edit nearly twice as efficient at position C18 (~60%) compared to C17 (~30%). Position C16 and C15 also show a considerable editing level (>10% base editing). Similar profiles were observed for VRER-Cas9, xCas9 and Cas9NG. Key characterizations of the work should be restated to reflect this.

Finally, the authors present their work in part as enabling more selective editing as a higher fraction of potential targeted bases, but they provide no quantitative estimate of the impact of this work. What fraction of disease-relevant Cs in the human exome are now addressable with this new set of reagents compared to previously existing reagents? In how many cases would the improved specificity profile be beneficial? Such an analysis would help readers gauge the impact and utility of this work.

Minor comments

In the introduction – the HDR pathway for editing is completely ignored. This should be remedied as this is the other major means of achieving discrete base changes.

First sentence of the second introductory paragraph says "As most of the mutations causing hereditary diseases in humans ... represent point mutations rather than loss-of-function mutations". This needs to be referenced.

Figure 1: Data visualization. Currently, the data from the BE carrying the full-length CDA1 and the nCDA1-BE3 is shown twice for comparison (black and grey bars). The PolyC-1-NGA, etc. could be combined in a single figure. Constant y-axis formatting over all experiments would also help to compare efficiency levels between the different reagents.

Figure 2: see Figure 1.

The truncated cytosine base editors are more specific due to the narrow base editing window and the therefore reduced bystander mutations. The authors describe this as more precise which might be confusing since higher precision is often used to describe increased targeting density.

C1, C2, ..., C20 nomenclature: This nomenclature does not match the nomenclature introduced in the first base editing papers: The Liu's lab position C1 is the author's position C20.

Page 15: "Illumina MiSeq 4000" platform. Miseq or HiSeq 4000?

Response to reviewers comments (with our response in red)

Reviewer #1:

Tan et al. Expanding the genome-targeting scope and the site selectivity of high-precision base editors.

This manuscript by Tan and colleagues presented the development of new high-precision BEs that expand the versatility of base editing in yeast. The authors mainly replaced the Sp-nCas9 sequence with the reported nCas9 variants which recognize different non-NGG PAMs. Meanwhile, they tested the six different deaminases by fusing nCas9 directly to their C-terminus and found A3A displayed the best performance. Lastly the authors made the guidelines for the choice of the optimal cytidine BE. In general, this work provided information for base editing, but it is not clear that this advance on its own meets the novelty standards for this journal.

Specific points,

1. Although the authors provided new base editors for the high-precision, all the nCas9 and deaminases have been reported in the previous research.

The Reviewer is correct in that the Cas9 variants and deaminases have been reported previously. The novelty of our manuscript lies in (i) the combination of these Cas9 variants (having different PAM recognition specificities) with deaminase truncations to confer high-precision base editing, thus substantially expanding the versatility of base editing while retaining the editing efficiency, and (ii) the discovery that A3A can be engineered to obtain high-precision base editors that preferentially edit positions that were previously inaccessible.

2. The authors tested few target sites, for example, only one target site for nCDA1- Δ 195-VQRBE3 and nCDA1- Δ 194-VQRBE3. It is not enough to get the solid conclusions.

We apologize for not having made this clear: For each base editor described in our manuscript, we tested at least two target sites. We first performed dideoxy chain termination sequencing of amplified PCR products (Supplementary Figures 1-4) and then selected the two best-performing base editors for each target for further analysis by deep sequencing. As reported previously (Tan, et al., 2019), the best-performing base editor for one target site is not necessarily the best base editor for another site. We, therefore, tested multiple truncated versions to obtain the best base editor(s) for each target site. This is now explained on p.6 of the revised manuscript.

3. The safety about base editing causes great attentions because of the off-target. The authors did not analyze the off-target for the new base editors.

The reviewer points out correctly that recently, base editors were reported to cause off-targets both in genomic DNA (Zuo et al., Science, 2019; Jin et al., Science, 2019) and RNA (Grünewald et al., Nature, 2019; Zhou et al., Nature, 2019) in an sgRNA-independent manner. Since these off-targets are caused by the deaminase, we exemplarily investigated the DNA off-target effects of three different deaminase fusions (nCDA1-BE3, cCDA1-BE3, nCDA1 Δ 190-BE3) and a BE-free control by whole genome sequencing. We found that the base editor with the CDA1 truncation exhibited reduced off-target effects compared to BEs with the full-length deaminase (new Fig. 4). The likely explanation is that a narrower editing window does not only result in higher editing precision, but also means lower probability of off-target editing. We believe that these are important findings, because they suggest that the design of BEs with improved editing precision also helps with addressing the off-target issue. In addition, although we did not test the RNA off-target editing of CDA1- and A3A-derived BEs directly, we show in our manuscript that the reported RNA off-target-eliminating mutation in A3A can be combined with a deaminase truncation (Suppl. Fig. 12) without any loss in activity or specificity. Moreover, compared with APOBEC1 and A3A, CDA1-based BEs are known to induce much less editing of RNA (Grünewald et al., Nat. Biotechnol, 2019). We describe the results of the off-target analysis on p.10-12 (and the methods involved on p.19-20) of the revised manuscript, and we also added a paragraph to the Discussion section explaining the significance of the findings (p.14-15).

4. BE3 is not fit for RCC-15D if R was A in Table 1.

We agree and have removed BE3 from the -15 row in Table 1.

5. More information about the influence factors on editing precision of BEs should be provided.

As suggested by the Reviewer, we have added information on known factors influencing the editing precision of BEs to the Discussion section of the revised manuscript (p.14).

Reviewer #2:

The manuscript authored by Tan et al reports the development of new cytosine base editors with diversified high-precision editing scopes and expanded PAMs in yeast. The authors first fused CDA1 truncations with nCas9 variants and obtained diverse cytosine base editors editing at C-18 precisely with expanded PAMs in addition to NGG. Additionally, the authors also tested the editing signatures of various APOBEC3A truncations fusing to nCas9, and obtained cytosine base editors editing at C-15/C-16 precisely. Generally, this manuscript provides new tools for C-T conversions with diversified editing scopes and expanded versatility, which are useful

for base editing applications. I have the following points to improve the manuscript:

1. The authors observed that CDA1 fusing to the C-terminus of SpCas9-NG(cCDA1-NGBE3) narrowed the editing window. However, they did not evaluate the editing signatures of CDA1 fusing to the C-terminus of other nCas9 variants including VQR-nCas9, VRER-nCas9 and nxCas9. Additionally, the authors should demonstrate whether CDA1 truncations fusing to the C-terminus of nCas9 variants would change the activity or editing window.

As suggested by the Reviewer, we performed additional experiments to evaluate the effects of CDA1 fusions to the C-terminus of other nCas9 variants on editing precision (new Supplementary Figures 5-6). We found that these fusions, albeit showing a narrower activity window than N-terminal fusions, do not reach the specificity of the best-performing fusions with CDA1 truncations. We have added the description of these data to the revised manuscript (p.7-8). As to the fusions to the C-terminus of nCas9, we have shown previously that CDA1 truncations fused to the C-terminus of nCas9 change neither the editing activity nor the editing window (Tan et al., 2019).

2. It has been reported that APOBEC3A induced significant off-target C-U conversions at RNA level, and specific mutations could eliminate those off-target activities at RNA level (e.g. Y130F, R128A)(Zhou et al. Nature 2019;Grunewald et al. NBT 2019). In consideration of the future applicability of newly established APOBEC3A truncations fusing to nCas9, it is important to examine whether APOBEC3A truncations still have off-target activities at RNA level, and whether established mutations (e.g. Y130F, R128A) affect the activities and editing windows of APOBEC3A truncations.

To address off-targets, we have conducted additional experiments to investigate the effects of two known deaminase mutations on the width of the base editing window and the BE activity when combined with proper A3A truncations (new Supplemental Figure 12). Introduction of the mutations into A3A (in A3A-BE3) did not reduce base editing efficiency, consistent with the previous report (Zhou et al., Nature 2019). It also did not narrow the base editing window. When we combined these mutations with the two optimal truncations (A3A Δ 186 and A3A Δ 182), we found that Y130F (but not R128A) in combination with the A3A version truncated at residue 186 (A3A(Y130F) Δ 186-BE3) displayed a base editing window and an editing efficiency that was similar to that of A3A Δ 186-BE3. These results and the associated methods are described on p.10-11 and p.18 of the revised manuscript. We also performed additional experiments to address off-target DNA editing (see response to Reviewer #1, point 3).

3. The authors demonstrated that hA3A-NL-BE3 had a broader editing window than A3A-BE3, which indicated the standard XTEN linker might narrow the editing

window. However, in Fig.2, the authors just compare A3A-BE3 (with XTEN linker) with A3Adelta-BE3 (without XTEN linker). The editing signatures of A3A-NL-BE3 and A3Adelta-BE3 with XTEN linker are needed for systematic comparisons.

Our strategy to narrow the editing window is based on the removal of non-essential sequences between the nucleoside deaminase domain and the nCas9 domain of the fusion protein, which likely results in more precise positioning of the deaminase domain on the target sequence. Although we observed that removal of the XTEN linker between the two domains leads to a slightly broader editing window, we also noticed that the editing peak shifted (i.e., A3A-BE3 showed the editing peak at C-15/C-16, whereas editing with A3A-NL-BE3 peaked at C-17/C-18; Supplementary Figure 10). The possible explanation is that linker removal somewhat alters the spatial structure of the fusion protein, thus affecting positioning of the deaminase domain on the target sequence. Several papers have reported similar effects on the editing window width of BEs by using the XTEN linker and other linkers of similar lengths (Kim, Y.B. et al., *Nat. Biotechnol.*, 2017; Tan et al., *Nat. Commun.*, 2019). This is now explained on p.9 of the revised ms.

Reviewer #3:

In a recent prior paper (*Nat. Commun.* 10, 439) the authors (Tan et al.) had shown that they could reduce the problem of on-target “bystander mutations” generated by Cas9 base editors by truncating their deaminase domain. This substantially narrowed the window of bases that were efficiently edited, in some cases to just a single preferentially edited base (e.g. base -18 relative to the PAM). In their current paper, the authors seek to extend this work to Cas9 variants developed by others to have altered or relaxed PAM requirements and therefore enhanced targeting range. They fuse six truncated CDA1 domains (AID homologue of sea lamprey) to 4 different Cas9 variants with relaxed PAM requirements to ultimately increase preference for editing position -18 and therefore reduce bystander mutations. All six truncated CDA1 domains have been described previously in the context of the author’s prior paper (*Nat. Commun.* 10, 439) and show a similar editing profile when fused to the SpCas9 variants used in this study.

The authors also screen truncation variants of alternative deaminase domains in an attempt to identify base editors having narrowed editing windows focused on a base different than -18. To accomplish this, they examine C-terminal truncations of cytosine deaminases such as AID, AICDA or APOBEC3A when fused to the native SpCas9. The authors identify three human APOBEC3A truncations with increased editing preference for target positions -15 and -16: $\Delta 190$, $\Delta 186$ and $\Delta 182$, albeit with considerably reduced efficiency (see Figure 2).

This work is of considerable potential interest to those who may wish to use base

editing for cellular engineering applications, as it focuses on the two main limitations of this technology (i) an inability to design canonical Sp Cas9 for efficient editing of many potential target bases, due to PAM restrictions, and (ii) the problem of editing additional bases besides the chosen one for any given application. However there are a number of considerations that should be addressed.

Major issues:

Many editing applications, and probably the large majority of therapeutic ones, will require highly efficient base editing (>80%) in relevant cell types, especially when used in multiplexing applications. A key concern of this reviewer is that the data presented in this work provides little support for the ability of the described proteins to enable such high efficiency editing in application-relevant settings. Even though all studies were carried out in yeast – which if anything should provide a less stringent system for gauging performance – levels of targeted editing rarely exceeded 60% with the new constructs. Given the recurrent problem in the field of Cas9 engineering of variant proteins exhibiting insufficient activity in application-relevant settings (e.g. see the authors' own comments on xCas9, and also an examination of high fidelity Cas9 variants in Nat Med 24 1216) it would seem reasonable to require demonstration by the authors that their new editors can perform efficiently at endogenous loci in human cells. Such a demonstration would also serve to show that the narrowed editing window that is crucial to the novelty of this work is preserved in more relevant settings.

This issue seems especially acute in studies of hAPOBEC3A truncations $\Delta 186$ and $\Delta 182$, which rarely exceeded 40% editing in their model target studies, and showed on-target editing efficiency is only ~10-17% at yeast Can1 locus. This makes it difficult to predict if high enough editing will be achievable in application-relevant settings. It also leaves open the question of how bystander mutations will behave if it is possible to increase modification levels by increasing dose.

We generally agree with the Reviewer on the requirement for highly efficient base editing for many therapeutic applications (although in at least some cases, editing would be done in isolated cell lines that then are introduced back into the patient). Current base editors, including APOBEC1-, CDA1- and A3A-derived BEs, do not (yet) reach such high efficiencies (Komor et al., Nature, 2016; Nishida et al., Science, 2016; Gehrke et al., 2018, Nat. Biotechnol). The focus of our current study was to increase precision of base editors, not necessarily efficiency. The series of high-precision base editors that we provide show largely unaltered base editing efficiency in comparison with previously described standard BEs. Although the current efficiency may not be high enough for many applications in *in vivo* gene therapy, current BEs will be useful in *ex situ* therapeutic applications and also applicable in other fields such as crop breeding (given that much of the useful genetic variation in plant breeding represent point mutations). We have addressed these issues in the Discussion section of the revised manuscript (p.16).

As to the low efficiency of the hAPOBEC3A-derived BEs at the *Can1* locus, please note that this frequency was determined by drug selection and, therefore, is lower than the frequencies determined by NGS, because the yeast strain is diploid (see Methods). Dosage effects were tested in our previous paper (Tan et al., Nat. Commun, 2019) by increasing the induction time, which however has not resulted in an increase of bystanders.

A second concern is that there is no examination of the cell-wide specificity behavior of the author's truncated deaminase variants. It is unclear how the Cas9-independent off-target profiles of CDA1 or hAPOBEC3A will be impacted by the different truncations on the DNA and RNA level. Establishing that the new variants retain suitable global specificity would seem to be an important requirement given the author's rationale for this work in part as providing improved therapeutic agents.

We agree that off-target effects are an important issue and have addressed them experimentally by whole-genome sequencing. For details, please see response to Reviewer #1, point 3. RNA off targets were addressed by testing mutations that are known to abolish RNA editing (see Reviewer #2, point 2).

There is also at times overstatement of the position-selectivity of the editors that have been developed. For example authors say that their truncated deaminases "specifically edit position C18 relative to the PAM sequence" but there is not enough evidence to support such a strong statement. As the authors themselves point out, nCDA1 Δ 188-VQRBE3 can only edit nearly twice as efficient at position C18 (~60%) compared to C17 (~30%). Position C16 and C15 also show a considerable editing level (>10% base editing). Similar profiles were observed for VRER-Cas9, xCas9 and Cas9NG. Key characterizations of the work should be restated to reflect this.

We agree that "specifically" is too strong, and have toned down the corresponding statements (using "preferentially" or "predominantly" instead).

Finally, the authors present their work in part as enabling more selective editing as a higher fraction of potential targeted bases, but they provide no quantitative estimate of the impact of this work. What fraction of disease-relevant Cs in the human exome are now addressable with this new set of reagents compared to previously existing reagents? In how many cases would the improved specificity profile be beneficial? Such an analysis would help readers gauge the impact and utility of this work.

In the revised ms, we describe two specific examples of human disease-causing mutations that now can be corrected with our new precision BEs: the early-onset Alzheimers disease-associated PSEN1-L166P mutation and the oculocutaneous albinism-causing TYR-Y327C mutation (p.13). Due to the lack of suitable bioinformatic tools, we have not been able to conduct a systematic survey of all disease-causing mutations. Together with the mention of multiple bystanders around

human disease-associated mutations in the Introduction section, we hope that these examples give at least an exemplary impression of the utility of our precision BEs.

Minor comments

In the introduction – the HDR pathway for editing is completely ignored. This should be remedied as this is the other major means of achieving discrete base changes.

We have added the following sentences to the Introduction: "Homology-directed repair (HDR) stimulated by DSBs can be used to introduce precise changes into target DNA sequences. However, HDR requires the presence of donor DNA as repair template and suffers from very low editing efficiency when correction of point mutations is attempted." (p.3).

First sentence of the second introductory paragraph says "As most of the mutations causing hereditary diseases in humans ... represent point mutations rather than loss-of-function mutations". This needs to be referenced.

References have been added, as suggested.

Figure 1: Data visualization. Currently, the data from the BE carrying the full-length CDA1 and the nCDA1-BE3 is shown twice for comparison (black and grey bars). The PolyC-1-NGA, etc. could be combined in a single figure. Constant y-axis formatting over all experiments would also help to compare efficiency levels between the different reagents.

We prefer not to combine these data in a single figure for two reasons: (i) the diagrams would become too large and too crowded, and (ii) the best-performing base editor for one target site is not the most suitable BE for another target site (which would make it difficult for the reader to make the proper comparisons).

Following the Reviewers suggestion, we now use constant y-axis formatting for all BEs addressing the same target site (revised Figures 1 and 2).

Figure 2: see Figure 1.

See above.

The truncated cytosine base editors are more specific due to the narrow base editing window and the therefore reduced bystander mutations. The authors describe this as more precise which might be confusing since higher precision is often used to describe increased targeting density.

We used the term "more precise", because it has been used in this context in several recent reports (e.g., Gehrke, J. M. et al. Nature Biotechnology, 2018; Tan et al.,

Nature Communications, 2019). To make clear what we mean by “precise”, we have added an explanation (and references; p.4 of the revised ms).

C1, C2, ..., C20 nomenclature: This nomenclature does not match the nomenclature introduced in the first base editing papers: The Liu’s lab position C1 is the author’s position C20.

Our nomenclature is consistent with that used in the first report of a CDA1-derived base editor (Nishida et al., Science, 2016), and was also used in our previous publication. We prefer this numbering of nucleotide positions, because it is unambiguous and defines the editing position more clearly (i.e., as distance from the PAM, thus making it invariant). If sgRNA length or spacer length change, the numbering of the nucleotides would also change in the alternative nomenclature (and C1 would no longer be the same nucleotide of the target sequence).

Page 15: “Illumina MiSeq 4000” platform. Miseq or HiSeq 4000?

We apologize for the typo. It has been corrected in the revised manuscript.

Reviewers' Comments:

Reviewer #2:

Remarks to the Author:

The authors have addressed my concerns and I have no other comments.

Reviewer #3:

Remarks to the Author:

The authors have addressed the majority of my concerns. However two responses exhibit shortcomings that I feel should be addressed to warrant publication. My original comments, the authors' response, and my followup are provided below:

Original review:

.. the authors present their work in part as enabling more selective editing as a higher fraction of potential targeted bases, but they provide no quantitative estimate of the impact of this work. What fraction of disease-relevant Cs in the human exome are now addressable with this new set of reagents compared to previously existing reagents? In how many cases would the improved specificity profile be beneficial? Such an analysis would help readers gauge the impact and utility of this work.

Author's response:

In the revised ms, we describe two specific examples of human disease- mutations that now can be corrected with our new precision BEs: the early-onset Alzheimers disease -associated PSEN1-L166P mutation and the oculocutaneous albinism-causing TYR-Y327C mutation (p.13). Due to the lack of suitable bioinformatic tools, we have not been able to conduct a systematic survey of all disease-causing mutations. Together with the mention of multiple bystanders around human disease-associated mutations in the Introduction section, we hope that these examples give at least an exemplary impression of the utility of our precision BEs.

My response:

Providing exemplars is really not a valid way to address this concern. Given the vastness of the genome finding two supportive examples doesn't rule out the potential for merely marginal improvements. Somewhat surprised by the author's lack of a comprehensive response due to "lack of suitable bioinformatics tools". Given the availability of full genome sequence, as well as lists of clinically relevant point mutations, the requisite analysis should require little more than some custom python code.

Original review:

In the introduction – the HDR pathway for editing is completely ignored. This should be remedied as this is the other major means of achieving discrete base changes.

Authors' response:

We have added the following sentences to the Introduction: "Homology-directed repair (HDR) stimulated by DSBs can be used to introduce precise changes into target DNA sequences. However, HDR requires the presence of donor DNA as repair template and suffers from very low editing efficiency when correction of point mutations is attempted." (p.3).

My response:

The literature simply does not support this statement. See e.g. Vakulskas et al., Nature Medicine

24 1216, who show up to 70% correction of the sickle allele in the therapeutically relevant cell type: CD34+ HSPCs cells (figure 6a). Would argue that for readers to properly understand and value the significance of this work, Tan et al. really need to provide a more credible treatment of HDR-mediated editing as a competing technology, which explicitly acknowledges the high efficiencies available via this method (citing Vakulskas would be a good way to do this...) but also highlights the potential advantages of a base editing approach such as lack of competition from competing repair pathways such as NHEJ.

Response to reviewers comments (with our response in red)

Reviewer #2:

No remaining criticisms.

Reviewer #3:

The authors have addressed the majority of my concerns. However two responses exhibit shortcomings that I feel should be addressed to warrant publication. My original comments, the authors response, and my followup are provided below:

Original review:

.. the authors present their work in part as enabling more selective editing as a higher fraction of potential targeted bases, but they provide no quantitative estimate of the impact of this work. What fraction of disease-relevant Cs in the human exome are now addressable with this new set of reagents compared to previously existing reagents? In how many cases would the improved specificity profile be beneficial? Such an analysis would help readers gauge the impact and utility of this work.

Authors response:

In the revised ms, we describe two specific examples of human disease- mutations that now can be corrected with our new precision BEs: the early-onset Alzheimers disease -associated PSEN1-L166P mutation and the oculocutaneous albinism-causing TYR-Y327C mutation (p.13). Due to the lack of suitable bioinformatic tools, we have not been able to conduct a systematic survey of all disease-causing mutations. Together with the mention of multiple bystanders around human disease-associated mutations in the Introduction section, we hope that these examples give at least an exemplary impression of the utility of our precision BEs.

My response:

Providing exemplars is really not a valid way to address this concern. Given the vastness of the genome finding two supportive examples doesn't rule out the potential for merely marginal improvements. Somewhat surprised by the author's lack of a comprehensive response due to "lack of suitable bioinformatics tools". Given the availability of full genome sequence, as well as lists of clinically relevant point mutations, the requisite analysis should require little more than some custom python code.

As suggested by the Reviewer, we have performed a systematic bioinformatic analysis of the ClinVar database for human disease-associated T-to-C (and A-to-G) SNPs (Supplementary Figure 14). As expected, addition of our new BEs to the available set results in a substantial increase in the disease-causing mutations that

become addressable (in that the expanded BE set roughly doubles the fraction of addressable mutations from 23% to 47%). The results and the associated methods are described on p.13 and p.19 of the revised manuscript (and illustrated in Supplementary Figure 14).

Original review:

In the introduction – the HDR pathway for editing is completely ignored. This should be remedied as this is the other major means of achieving discrete base changes.

Authors response:

We have added the following sentences to the Introduction: "Homology-directed repair (HDR) stimulated by DSBs can be used to introduce precise changes into target DNA sequences. However, HDR requires the presence of donor DNA as repair template and suffers from very low editing efficiency when correction of point mutations is attempted." (p.3).

My response:

The literature simply does not support this statement. See e.g. Vakulskas et al., Nature Medicine 24 1216, who show up to 70% correction of the sickle allele in the therapeutically relevant cell type: CD34+ HSPCs cells (figure 6a). Would argue that for readers to properly understand and value the significance of this work, Tan et al. really need to provide a more credible treatment of HDR-mediated editing as a competing technology, which explicitly acknowledges the high efficiencies available via this method (citing Vakulskas would be a good way to do this...) but also highlights the potential advantages of a base editing approach such as lack of competition from competing repair pathways such as NHEJ.

We agree and have modified the statements according to the suggestions of the Reviewer (also citing the suggested reference; p.3).